# Photonic Crystal Nanobeam Cavities for Nanoscale Optical Sensing: A Review

**DOI:** 10.3390/mi11010072

**Published:** 2020-01-09

**Authors:** Da-Quan Yang, Bing Duan, Xiao Liu, Ai-Qiang Wang, Xiao-Gang Li, Yue-Feng Ji

**Affiliations:** State Key Laboratory of Information Photonics and Optical Communications, and School of Information and Communication Engineering, Beijing University of Posts and Telecommunications, Beijing 100876, China; ydq@bupt.edu.cn (D.-Q.Y.); duanbing@bupt.edu.cn (B.D.); liuxiao0921@bupt.edu.cn (X.L.); waq@bupt.edu.cn (A.-Q.W.); Lixiaogang@bupt.edu.cn (X.-G.L.)

**Keywords:** photonic crystal nanobeam cavity, optical sensor, sensing mechanisms, refractive index sensing, nanoparticle detection, biomolecule detection, multiplexed sensing

## Abstract

The ability to detect nanoscale objects is particular crucial for a wide range of applications, such as environmental protection, early-stage disease diagnosis and drug discovery. Photonic crystal nanobeam cavity (PCNC) sensors have attracted great attention due to high-quality factors and small-mode volumes (Q/V) and good on-chip integrability with optical waveguides/circuits. In this review, we focus on nanoscale optical sensing based on PCNC sensors, including ultrahigh figure of merit (FOM) sensing, single nanoparticle trapping, label-free molecule detection and an integrated sensor array for multiplexed sensing. We believe that the PCNC sensors featuring ultracompact footprint, high monolithic integration capability, fast response and ultrahigh sensitivity sensing ability, etc., will provide a promising platform for further developing lab-on-a-chip devices for biosensing and other functionalities.

## 1. Introduction

Ultra-sensitive and rapid detection of nanoscale analytes plays an important part in numerous fields, such as homeland security, environmental protection, early stage disease diagnosis and drug discovery [1,2,3]. For example, the detection of nanoparticles in air is important for human health, because the ultra-fine particles can destroy lungs, leading to lung disease. Moreover, they also have the ability to penetrate membrane cells and diffuse along the synapses, blood vessels and lymphatic vessels of nerve cells resulting in serious illness [4,5,6,7,8,9,10]. The monitoring of fatal infectious bacteria and viruses is necessary for early-stage disease control, including severe acute respiratory syndrome (SARS) and acquired immune deficiency syndrome (AIDS) [11,12]. 

Enabled by the rapid development of nanofabrication technology, methods such as fluorescent labelling and enzyme-linked immunosorbent assays (ELISA) have been developed for detecting nanoscale objects and the detection abilities is down to single nanoparticle level. However, labeling the target analytes always needs complicated processes and might affect their chemical structures. Label-free optical sensors exhibit distinguished potential for the detection of nanoscale analytes (physical, chemical, and biological) owing to their good performance in non-invasiveness, fast response, real time in situ detection capability and immunity to electromagnetic interference. In recent years, various optical sensors have been developed for sensing applications, including mach-zehnder interferometers (MZI) [13,14], surface plasmon resonators (SPR) [15,16], fiber Bragg gratings (FBG) [17], and optical microcavities [18,19]. Among these, optical microcavities are considered to be excellent platforms for optical sensing due to the high-quality factors (Q) and small-mode volumes (V). The optical microcavities confine the light in a small volume, leading to significant enhancement of light–matter interaction, resulting in ultra-high sensitivity (S) and a low detection limit. When subject to slight environmental changes, the spectral properties change can be obtained, e.g., resonator wavelength shift [20,21,22] and mode broadening [23].

The main types of optical microcavities include whispering gallery mode (WGM) cavities, Fabry–Perot (F–P) cavities and photonic crystal (PhC) cavities [24]. Among these, PhC cavities have been investigated as advantageous platforms due to their ultra-high Q/V enabling the enhancement of light–matter interactions. Particularly, compared with two-dimensional (2D) PhC cavities, one-dimensional photonic crystal nanobeam cavities (1D-PCNCs) attract great attention for their ultra-sensitive optical sensing and lab-on-a-chip applications, owing to their ultra-compact size, ultra-small mode volume, high integrability with bus-waveguide and excellent complementary metal-oxide-semiconductor (CMOS) compatible properties [25,26,27,28,29,30,31,32,33]. 

In this review, we will focus on nanoscale optical sensing based on 1D-PCNCs. The structure of the review is organized as follows. A comprehensive overview about basic properties and sensing mechanisms of 1D-PCNCs sensors is discussed in Section 2. In Section 3, firstly, we introduce the efforts to pursue ultra-high figure of merit (FOM) in refractive index (RI) sensing based on 1D-PCNCs. Secondly, we outline the applications of 1D-PCNC sensors on single nanoparticle and label-free molecule (viruses, proteins and DNAs) detection. Thirdly, a monolithic integrated 1D-PCNC sensor array for multiplexed sensing is introduced explicitly. In addition, we present other sensing applications of 1D-PCNC sensors such as temperature sensing and optomechanical sensing, etc. Next, we review the nanofabrication and coupling techniques in the development of 1D-PCNC sensors in Section 4. Finally, we draw a brief conclusion.

## 2. Sensing Mechanisms of Photonic Crystal Nanobeam Cavities (PCNC) Sensors

The resonance characteristics, such as resonant wavelength (λ) and linewidth (δλ), could be notably influenced by tiny physical, biological and chemical changes in the optical mode region. With recent advances in research on PhC cavities both theoretically and experimentally [34,35,36,37,38], PCNCs have been extensively applied for optical sensing. PCNC sensors are advantageous for optical sensing because of their ultra-high Q/V. Firstly, Q can be expressed as the ratio of the cavity central resonance wavelength to its linewidth, namely, Q = λ_0_/δλ = ωτ, where ω is the angular frequency and τ is the cavity photon life time. The longer the photon life time, the stronger the interaction between light and matter will be. Higher Q exhibits narrower resonance linewidth (δλ), and determines the minimum resonance shift (∆λ) detectable by the system. Next, V describes the light confinement ability. A small V implies that the mode field is highly localized, which can significantly enhance the light–matter interactions. The high Q and small V enable 1D-PCNC to be an excellent platform for lab-on-a-chip optical sensing based on the resonance mode shift or mode broadening mechanism [19].

### 2.1. Mode Shift

The common sensing mechanism based on PCNC is to monitor the resonance wavelength shift on account of analytes-induced RI variations in the near-field of the PCNC. These variations can be put down to local perturbations induced by the adsorption of analytes onto the cavity surface (e.g., adsorption of proteins), or global perturbations in the background RI induced a variation in the bulk concentration of the analytes (e.g., the different concentrations of sodium chloride solutions). According to perturbation theory [39], the wavelength shift Δλ generated by a small index perturbation Δn of the surrounding analytes can be calculated. Taking into consideration of a homogenous change of refractive index, we can assume that the changes of RI is all the same in the perturbed region and there is no change in the unperturbed region. Finally, the wavelength offset can be approximated as:(1)$$\frac{\Delta \lambda }{{\;\lambda }_{0}}\approx \frac{\Delta n}{n_{0}}\frac{\int_{V_{\mathrm{analyte}}}\varepsilon \left(r\right){\left|E\left(r\right)\right|}^{2}d^{3}r}{\int_{V_{\mathrm{total}}}\varepsilon \left(r\right){\left|E\left(r\right)\right|}^{2}d^{3}r}=\frac{\Delta n}{n_{0}}{\;\Gamma }_{\mathrm{analyte}}$$
where $\Gamma_{\mathrm{analyte}}$ describes the filling factor of the electric field energy stored in the surrounding analytes. Furthermore, when the PCNC is used for nanoparticles or molecular detection, ∆λ of the resonance can be described as follows [21]:(2)$$\frac{\Delta \lambda }{\lambda_{0}}\;=\frac{3\left(\varepsilon_{p}-\varepsilon_{s}\right)}{\varepsilon_{p}+2\varepsilon_{s}}\;\frac{{\left|E_{\mathrm{mol}}\right|}^{2}}{2\int \;\varepsilon {\left|E\right|}^{2}\mathrm{dr}}V_{\mathrm{mol}}$$
where $\varepsilon_{p}$ is the permittivity of the nanoparticle, $\varepsilon_{s}$ is the permittivity of the background environment, E_mol_ is the electric field at the nanoparticle location, $\int \;\varepsilon {\left|E\right|}^{2}\mathrm{dr}$ is the overall optical mode energy inside cavity, and $V_{\mathrm{mol}}$ is the nanoparticle volume. From Equations (1) and (2), it is obviously presented that a smaller cavity V and a stronger light confinement in the perturbed region can lead to a larger ∆λ and then result in higher sensitivity. Here, the sensitivity is defined as S = $\;\frac{\Delta \lambda }{\Delta n}$. According to Equation (1), the S of the cavity could be expressed as [39]: (3)$$S=\frac{\Delta \lambda }{\Delta n}\approx \frac{\lambda_{0}}{n_{0}}\Gamma_{\mathrm{analyte}}$$
indicating that the S relies much on the $\Gamma_{\mathrm{analyte}}$. High $\Gamma_{\mathrm{analyte}}$ would lead to a high S, because the light field can fully overlap with the analytes in the low-index region. 

Herein, we can define the FOM of PCNC sensors as [40]:
FOM = Q·S/λ_0_(4)
where λ_0_ is the cavity resonant wavelength. 

### 2.2. Mode Broadening

Recently, PCNC sensors based on the dissipative interaction have reduced the detection limit to single nanoparticle level. In the case of a PCNC with an absorptive material surrounding, the absorption causes the cavity mode linewidth broadening. There are no requirements for the ultra-high Q and it is insensitive to external disturbance such as environmental temperature drift and the probe laser in the cavity mode broadening measurement. Compared with mode shift, a mode broadening sensing mechanism is determined by the dissipative characteristics of the nanoparticles and is well suitable for high-Q, low index-contrast polymeric photonic crystal nanobeam cavities to detect lossy analytes. The total cavity Q can be written as [41]:(5)$$\frac{1}{Q_{\mathrm{total}}}\;=\;\frac{1}{Q_{r}}\;+\frac{1}{Q_{c}}+\Gamma_{\mathrm{analyte}}\frac{1}{Q_{\mathrm{abs}}}$$
where 1/Q_r_, 1/Q_c_, 1/Q_abs_ represent the radiation loss, the input/output waveguides-induced loss, the material absorption loss, respectively. To calculate Q_abs_, the filling factor $\Gamma_{\mathrm{analyte}}$ should be calculated first (in COMSOL Multiphysics [42]). The $\Gamma_{\mathrm{analyte}}$ and Q_abs_ can be written as [41]:(6)$$\frac{1}{Q_{\mathrm{abs}}}=2\Gamma_{\mathrm{analyte}}\frac{k}{n}$$
where k and n are the imaginary and real parts of the complex RI ($\tilde{n}=n+\mathrm{ik}$) of the absorptive material. For a given PCNC structure, the filling factor $\Gamma_{\mathrm{analyte}}$ relies on the real part of the surrounding analytes. When detecting the analytes, 1/Q_r_ and 1/ Q_c_ hardly changes, so we can express it as [43]:(7)$$\Delta \left[\frac{1}{Q_{\mathrm{total}}}\right]=\Delta \left[\frac{\delta \lambda }{\lambda_{0}}\right]=\frac{\Delta \left[\delta \lambda \right]}{\lambda_{0}}-\frac{\delta \lambda \cdot \lambda }{\lambda_{0}^{2}}=\Delta \left[\Gamma_{\mathrm{analyte}}\frac{2k}{n}\right]$$
where $\lambda_{0}$ is original resonant wavelength, and δλ is optical mode linewidth. Apparently, the change of δλ is linearly proportional to the change of k. Therefore, the analytes–sensor interaction is translated into the change of the k, and then converted into the changes of the Δ[δλ]. Note that the side scattering caused by the analytes also leads to the resonance energy loss, resulting in the linewidth broadening and Q drops [44], which have been considered as having huge potential for detecting single nanoparticles, therefore possibly providing an excellent platform for practical demands in single molecule detections.

## 3. Sensing Applications of PCNC Sensors

### 3.1. Efforts on Ultra-High Figure of Merit (FOM) for Refractive Index (RI) Sensing

As introduced in Section 2 above, the FOM of PCNC sensors is expressed as FOM = S·Q/λ_res_ [40]. However, FOM is affected by the trade-off between S and Q: in order to obtain high S [45,46], the light mode requires strong interaction with the analytes (i.e., outside the waveguide medium); in order to achieve high Q, the light mode should be more confined to the waveguide medium. As a result, the best geometry to maximize Q and S is being actively developed, which provides excellent platform for achieving ultra-low detection limit of single nanoparticle and biomolecules detection. Here, Table 1 summarizes a wide range of comparisons between different structures.

For higher Q, the light mode should be more confined to the waveguide medium, Rahman et al. presented the design modelling and fabrication of silicon-on-insulator (SOI) nanobeam cavities, which was immerged in a microchannel for RI sensing [47]. Sensitivity of this sensor was more than 200 nm/RIU (refractive index unit) while Q exceeds 30,000 in air and 20,000 in water. Then, Gong et al. [48] designed and fabricated 1D-PCNCs using silica. They used trapezoidal cavity designs in Si and Si_3_N_4_ [27,36], but considered silica plates suspended in free space. The simulation showed that Q exceeded 10^4^ with the mode volume of 2.0 (λ/n)^3^ at the visible wavelength range (600–716 nm). For example, Yao et al. reported an optical sensor with an ultra-high sensitivity based on the 1D-PhC stack mode-gap cavity [49]. By introducing the width-tapering structure that was quadratically modulated, a waveguide coupled 1D-PhC stack mode-gap cavity was designed. The measured Q was up to 1.74 × 10^7^ with mode volume of 1.48 (λ/n_Si_)^3^ [49]. The confined mode accounted for most electric field energy, which interacted strongly with the detection target. The S was as high as 269 nm/RIU.

For a higher sensitivity, the optical resonance field requires to be confined in a small mode volume and interact strongly with the analytes. The S can be improved by the following methods: designing the air mode (AM) PCNC to localize the optical field in the air holes (circular [50,51], elliptical [52]); embedding a nano-slot into the dielectric mode (DM) PCNC and the light can be localized in the nano-slot [40,53,54,55,56,57,58]. For example, Wang et al. presented InGaAsP PhC slot nanobeam slow optical waveguides, which embedded InAs quantum dots [53]. The 900 nm/RIU measurement record of S fully confirms that the electric field can be strongly localized in the slot region. In 2017, Liu et al. proposed a suspended ultra-sensitive silicon nitride (Si_3_N_4_) PCNC sensor [59]. Si_3_N_4_ has been considered as an alternative material for SOI platforms because of low-cost, CMOS compatible properties and relatively low RI (n ~ 2). Furthermore, the loss of Si_3_N_4_ waveguides based on plasma-enhanced chemical vapor deposition (PECVD) thin films at the visible wavelength band was less than 1 dB/cm [60]. All outstanding advantages introduced above enable Si_3_N_4_ to be a reliable choice for visible sensing applications. The S of 321 nm/RIU was also obtained at the wavelength of ~700 nm with higher order mode. Moreover, the slotted geometry was exploited to provide strong light confinement and enhanced localized field, which resulted in an ultra-small mode volume. Thus, the performance advantages of ultra-high sensitivity, ultra-high Q factor, and ultra-small mode volume enabled the proposed single 1-D PhC-SNCs to be promising candidates for lab-on-chip sensing arrays.

Much research has been proposed to optimize structure design to obtain high Q and high S simultaneously with PCNC RI sensors. For example, Quan et al. presented the design of high Q polymeric PCNCs in 2011, which consisted of two polymers with the refractive index contrast of 1.15 [61]. Furthermore, at a 100 microwatt power level, thermo-optical bistability could be observed. The nanobeam cavity included a ridge waveguide, and the ridge waveguide was perforated with elliptical holes grating. At the same time, the cavities immersed in D_2_O was measured, and the absorption of the cavity in the range of telecommunication frequency was negligible. The highest Q factor obtained in D_2_O was 36,000. A lower ratio of the system (index ratio = 1.15) was achieved while the Q was almost one order of magnitude higher than that formerly proved (Q = 5000 in air and index ratio = 1.46) [36]. They also demonstrated that the FOM of these devices was two orders of magnitude higher than sensors based on SPR, and better than the commercial Biacore^TM^ sensors. The evanescent field of the polymer nanobeam cavity can be extended to the surrounding medium, which is more than the widely applied silicon-based nanophotonic sensors because of the low RI contrast. Therefore, the wavelength shift of the polymer cavity was more sensitive to the variations of the volume RI in the ambient medium. By combining functional nanostructures with new materials, we hope to inspire new applications such as non-linear optics, light-emitting diodes (LEDs), and photoluminescent devices. Moreover, Yang et al. demonstrated a 1D-PhC single nanobeam cavity based on air mode in 2015 [50]. For the AM, the optical resonance field was strongly confined in the air area and overlapped with the analytes, enhancing the interaction between light and matter. Consequently, a high Q of 5.16 × 10^6^ and S of 537.8 nm/RIU were achieved at wavelength of 1350 nm. In most sensing applications where water was used as carrier liquid, the Q of sensor has been limited to about 10^4^ due to the absorption of water at the telecommunication wavelength, resulting in a FOM of about 4000. Therefore, the 1D-PhC single nanobeam AM cavity sensor was considered a promising platform for RI sensing. In addition, a nano-slotted parallel multibeam cavity was proposed by Yang et al. in 2013. When absorption was ignored, they obtained S > 800 nm/RIU and Q > 10^7^ in liquid in the range of telecommunication wavelength [62]. Then, Kim et al. demonstrated a high Q and S optical sensor based on PCNC with the second lowest air band edge mode, achieving S of ~631 nm/RIU and Q > 23,300 and theoretical FOM was more than 9500 [63]. In order to achieve ultra-high Q and S simultaneously, Yang et al. proposed two different designs (AM, DM) of ultra-compact photonic crystal nanofiber cavities (PCNFCs) in 2018. By numerical simulation, the Q and S of Q ~ 1.1 × 10^7^, S = 563.6 nm/RIU and Q ~ 2.1 × 10^5^, S = 736.8 nm/RIU were obtained, and the FOM was up to 4.31 × 10^6^ and 1.13 × 10^5^, respectively [64]. Compared with the previous works of PCNFCs, Q/V was more than two orders of magnitude. In addition, the index contrast of proposed device in this work was reduced more two times than previous research works. In 2017, Sun et al. proposed a single PhC slot nanobeam cavity with rectangular-shaped air holes, which achieved S of 835 nm/RIU and Q of 5.50 × 10^5^ with mode volume down to 0.03 (λ/n_air_)^3^, and an ultra-high FOM with 2.92 × 10^5^ was obtained [65]. Obviously, the light was mainly confined in the slotted region which indicated analytes can be strongly interacted with light. Xu et al. designed high Q slotted 1D-PhC cavities with parabolic-width stack in 2013, achieving Q of 3.73 × 10^6^ with small mode volume of 0.217 (λ/n_c_)^3^ at λ = 1546.5 nm [66]. Both a high S of 437 nm/RIU and FOM of 1.5 × 10^6^ was obtained. Later, Yang et al. designed a single 1D-PhC slot nanobeam cavity modulated by parabolic radius in 2015, which was achieved by introducing a nanoslot in the middle of a 1D-PCNC and was composed of air-holes (n_air_ = 1.0) with reduced radii and etched into a silicon (n_si_ = 3.46) ridge waveguide. A high Q of about 4.10 × 10^3^ and a transmission of nearly 70% can be acquired from the simulation. The fundamental mode wavelength was 1529.92 nm. In addition, they demonstrated that both an ultra-high S of 750.89 nm/RIU and Q of 2.67 × 10^7^ in air can be obtained simultaneously, resulting in FOM > 10^7^ [67]. The results demonstrated that the proposed sensor is potentially an ideal platform for on-chip sensing owing to high sensitivity, small footprints and masses.

In addition, PCNCs with ultra-high sensitivity, quality factor and ultra-small mode volume opens up the possibility to push the detection limit down to single nanoparticle level [44]. Furthermore, the sensitivity and detection limit can be enhanced by introducing SPR into PCNCs in the future, which has been employed in WGM cavities [68]

### 3.2. Single Nanoparticle Trapping and Detection

Detection of a single nanoparticle is of enormous importance in early-stage disease diagnosis, explosive materials detection and semiconductor manufacturing. Optical sensors such as nano-waveguide sensors [69,70,71,72], WGM microcavity sensors [73], and PhC microcavity sensors [74,75] have attracted great attention due to high S, low cost and small size, which has been successfully used for single nanoparticle detection. Moreover, the sensitivity and detection limit are the major factors to characterize the performance of optical sensors. Compared with nano-waveguide sensors, optical microcavities can confine light in a small mode volume, which results in the strong light–environment interactions and then leads to ultra-high sensitivity as well as the low detection limit. Among these, PhC cavities have potentially higher sensitivities than WGM cavities due to their small cavity volume [44]. Therefore, the detection limit is comparable to WGM cavities although the Q of PhC cavities is not as high as WGM microcavity sensors. In addition, the CMOS compatible property of PhC cavities opens up the possibility of integrate PhC devices for sensing. In the recent years, efforts on pursuing ultra-low detection limit and high sensitivity have been carried both in air and aqueous environment by designing different structures and different methods of nanoparticle capture. Some examples are summarized in Table 2, providing an overview about the ability of single nanoparticle detection using PCNCs. For instance, Lin et al. trapped and detected particles with diameter of 110 nm using optical forces in the year of 2012, which avoided the diffusion bottleneck and enhanced the reusability [76]. In 2014, Lin et al. proposed a PhC waveguide cavity with a waist structure for a 50 nm radius nanoparticle trapped on the lateral side of the waist and the device presented advantages of ultra-compact, efficient and versatile, contributing to the development of lab-on-chip sensing [77]. Lin, et al. proposed a surface-enhanced Raman scattering platform based on PCNC, achieving the detection of an Ag nanoparticle with radius down to 40 nm in 2013 [74]. The nanoparticle can be released and trapped by turning the laser off and on, which shows an excellent reusability and reconfigurability. Another structure was demonstrated with a slot between two neighboring holes to confine light by Wang et al. in 2015, detecting single polystyrene (PS) nanoparticles with radii of 20nm and 30nm in aqueous environment [58]. In the year of 2013, Quan et al. fabricated PCNC in the SOI platform with Q of 1.2 × 10^5^, 8.3 × 10^4^, 3.5 × 10^4^ in air, D_2_O, H_2_O and detected polystyrene nanoparticle with the lowest radius of 12.5 nm in DI water [21]. Yang et al. theoretically showed that a PS nanoparticle as small as 10 nm in radius can be trapped and detected in 2018 [78]. A 1D-PhC microcavity biosensor with Q of 190 was presented by Schmidt et al. in 2004, achieving the detection of gold particles 10 nm in diameter, showing the ability of biomolecules detection, such as DNA, RNA, proteins and antigens [79]. Smaller nanoparticle with radii of 1 nm were detected theoretically by Lin et al. in the year of 2015 [55]. In the same year, Liang et al. experimentally demonstrated the first air-mode PCNCs at telecom wavelength with high Q of 2.5 × 10^5^, low-mode volume of 0.01 (λ/n_air_)^3^. In this configuration, nanoparticles deposited by electrospray and smaller single gold nanoparticles with diameter of 25 nm, 15 nm, 5 nm and 1.8 nm were detected based on mode shift and mode broadening simultaneously [44]. The combination between PCNC and piezospray or electrospray provided a research method to characterize limited nanoparticle size frequently and rapidly.

### 3.3. Biomolecules Detection

Biosensing plays a significant part in disease diagnosis, such as the detection of different molecules in a blood sample, which is always based on an antibody–antigen locking mechanism. Hence, researches have been put forward to optimize PCNCs design in order to obtain high FOM and ultra-low detection limit for biomolecules detection. Several research projects have been carried out for biosening by integrating microfluidic with PCNCs. For example, by embedding PCNCs inside a microfluidic channel and functionalized with biotin, as shown in Figure 1a, single streptavidin molecule with a concentration of 2 pM was detected, as shown in Figure 1b [21]. This device provided a platform for ultra-sensitive and label-free biomolecules detection. Tardif et al. finished the identification of three different kinds of bacteria using 1D-PhC, which provided a reference for the investigation of larger numbers of bacteria with different shapes, sizes and properties, as shown in Figure 1c,d [80]. Based on PCNCs, Liang et al proposed PhC chips with Q of 14,000 for label-free protein detection and detected carcinoembryonic antigen (CEA) biomarker from 0.1 pg/mL to 10 μg/mL [81], as shown in Figure 2a,b. In addition, the device can be mass-produced with scalable deep-ultraviolet (UV) lithography, enabling biological sensor chips to be practical and commercial. To obtain high Q and high S simultaneously, Yang et al. proposed nanoslotted parallel quadrabeam PhC cavity (NPQC) with high FOM > 2000, as shown in Figure 2c and experimentally observed streptavidin-biotin binding events via mode shift, achieving the detection of streptavidin molecule with concentration down to 10 ag/mL, as shown in Figure 2d [54]. Later, in the year of 2017, Liang et al. designed a hybrid photonic-plasmonic antenna-in-a-nanocavity biosensor, as demonstrated in Figure 3a [82]. Furthermore, they have successfully achieved the label-free detection of DNA-protein dynamics with an ultra-high Q/V, as shown in Figure 3b. The research demonstrated the limitation of traditional fluorescent labeling methods and provide an attractive option for single-molecule drug discovery. Rodriguez et al. demonstrated a PCNC by utilizing a porous silicon with large internal surface area, which allowed enhanced molecular capture and direct light–matter interaction in 2019 [83], as shown in Figure 3c. In this work, the surface detection S was approximately 1.6 pm/nM for 16-base nucleic acid molecules, as depicted in Figure 3d.

In summary, for both the detection of single nanoparticles and biomolecules introduced above, the adsorption of single nanoparticles and biomolecules onto the PCNCs is monitored via the resonance wavelength shift and linewidth broadening. Many methods have been proposed to pursue high FOM and low detection limit, such as studying the functionalized materials to reduce absorption loss, making the functionalization process easy to operate, and improving persistence and optimizing the structure of nanobeam cavities.

### 3.4. Monothlic Integrated Sensor Array for Multiplexed Sensing 

With the rapid development of micro-nano integrated devices, higher requirements are placed on miniaturization, functional diversification and high integration. Therefore, there is much research work focused on multiplexed sensors including nanobeam cavities array and single nanobeam cavity, which have realized the detection of various analytes.

Mandal et al. proposed a nanoscale multi-channel optofluidic sensor array based on PCNCs, achieving the label free detection of biomolecules in an aqueous environment in 2008 [84]. In this work, PCNCs side coupled to input/output waveguides, and each nanobeam cavity and waveguide was separated in different fluid channels, as shown in Figure 4a. To illustrate the nanoscale optofluidic sensor array, the authors fabricated fluidic architecture by a soft lithography technique with a flexible polymer PDMS, which successfully achieved the detection of water and CaCl_2_ solution with a RI resolution of 7 × 10^−5^ and the detection limit of 35 ag. Later, they demonstrated the multiplexed ability of the structure introduced above via the separate and simultaneous detection of three kinds of interleukins in 2009 [85], as shown in Figure 4b. Moreover, Yang et al. demonstrated a 3-parallel-channel integrated sensor array with footprint of 4.5 μm × 50 μm (width × length) in 2016 and the S of each channel is 112.6 nm/RIU, 121.7 nm/RIU, 148.5 nm/RIU [86], shown in Figure 4c,d. These ultra-compact devices have been considered as promising platform for further lab-on-chip applications with high integration and ultra-sensitive multiplexed sensing abilities. Except for a nanobeam sensors array, a single nanobeam cavity also can be used for multiplexed sensing. For example, another cascaded side coupled structure for multiplexed sensing has been proposed by Liu et al. [87], as shown in Figure 4e. There two PCNCs for different sensing, one is air-mode cavity for RI sensing and another is dielectric-mode cavity for temperature sensing. Moreover, experiments have been carried out for the RI sensing and temperature sensing in glucose solution, obtaining the highest RI sensitivity of 254.6 nm/RIU for air-mode cavity and temperature sensitivity of 56.4 pm/°C for dielectric-mode cavity, as shown in Figure 4f. This structure shows the excellent capabilities in multiparametric sensing applications due to the simple configuration, easy fabrication and convenient integration.

In general, with the rapid development of PCNC design and fabrication over the past few years, PCNCs have been considered an excellent platform for nanoscale ultra-compact integrated sensor array devices with ultra-high S and multiplexed sensing abilities owing to their ultra-small Q/V, as well as high integrability with waveguides and photonic circuits [88].

### 3.5. Other Applications

Except for the applications of PCNCs in RI sensors, biosensors, and multiplexed sensors, other sensing applications of the PCNCs also have been proposed, such as temperature sensing and optomechanical sensing.

Temperature sensors have been widely used in many fields, like automobiles [89], environmental applications [90], health care [91], and manufacturing [92]. For example, in 2016, Zhang et al. demonstrated an ultra-sensitive temperature sensor with cascaded silicon PCNCs, as illustrated in Figure 5a. To increase the S, they designed two PCNCs; one was the stack width modulated structure with the light mode mainly localized in the SU-8 region, while another was parabolic-beam structure with the light mode strongly localized in silicon core, as depicted in Figure 5b,c. By combining the silicon with SU-8, the S was greatly improved because of the opposite thermo-optic coefficients between them, i.e. positive for silicon and negative for SU-8. Because the parabolic-beam cavity experienced a red shift while stack width modulated cavity experienced a blue shift with the temperature increased, the S measured experimentally of the sensor was around 162.9 pm/°C, almost twice as high as the conventional sensors based on silicon [93]. Since the sensitivity is enhanced, the proposed temperature sensor can be potentially used for an on-chip system combining with the integrated low resolution micro optical spectrometry such as an arrayed waveguide grating. In addition, the work also held the potential for high-precision sensing by compensating thermal refraction effect based on two materials with opposite thermo-optic effect.

PCNCs can also be applied for physical sensing by adjusting the coupling between a light field and nanomechanical motion. Moreover, 1D-PCNCs have attracted great attention in optomechanical sensing owing to their smaller size as well as the higher design flexibility. For example, Wu et al. in 2014 proposed a physical sensor based on a PhC split-beam cavity, achieving the detection of optomechanical torque. The measurement of sub-pg torsional can be realized, obtaining the S of 500 MHz/nm for dissipative coupling and 2 GHz/nm for dispersive coupling. In addition, as shown in Figure 6a, the S of the cantilever-type mechanical resonance sensor was 1.2 × 10^−20^ nm/√Hz and 1.3 × 10^−21^ nm/√Hz under environmental conditions and low vacuum, respectively [94]. In 2015, Kaviani et al. introduced an optomechanical sensor with strong non-linear optomechanical coupling, low mass, and wide light mode spacing [95]. As illustrated in Figure 6b, it was a photonic crystal “paddle nanocavity” that combined working principles of membrane-in-the middle (MIM) cavities [96,97] and PCNC optomechanical devices [98]. Leijssen et al. [99] then realized an optomechanical sensor using a sliced PhC nanobeam, combining such strongly localized light fields with a low-mass mechanical mode, as shown in Figure 6c. Eichenfield et al. [27] demonstrated an optomechanical sensor consist of two Si_3_N_4_ nanobeams in the near-field of each other, forming a so-called “zipper” cavity. In 2017, Du et al. introduced a resonant Lorentz force magnetic-field sensor based on dual-coupled PCNCs [100], as shown in Figure 6d. Compared with the Lorentz force magnetometer of microelectro mechanical systems (MEMS), the magnetic-field sensor presented the advantages of ultra-compact and a wide working bandwidth (160 Hz). The sensor has a much smaller footprint and thus have a potentially higher spatial sensing resolution of the magnetic field. It may be suitable for magnetometer arrays employed for a magnetic marker.

Gas detection is indispensable in many fields such as medical, automation as well as environment monitoring. The working principle of gas sensors is mainly based on conductance, capacitance, temperature detections [101,102] influenced by gas. For example, Feng et al. proposed the use of PCNC to obtain an ultra-compact, low-cost, and high S spectroscopic gas sensor in 2012. The whole spectroscopic gas sensor was composed of the radiation source, the interaction part and the detector [103], as illustrated in Figure 7a. The PCNC contained seven tapered air holes and 10 mirror holes, as shown in Figure 7b. This modeling achieved Q of 10^6^ and obtained 0.19 nm resonance wavelength shift by 10^−3^ RI change. With increase of the Q factor, the spectral resolution requirement and cost would be reduced while the sensitivity of the gas sensor would be improved. It is believed that PCNC will provide a promising application prospect for optical gas sensor.

In 2014, Chen et al. proposed a CMOS-compatible PCNC, which was modified with a fluoroalcohol polysiloxanes polymer, achieving the detection of the MeS concentration in the gas phase [104]. As shown in Figure 7d, the bus waveguide with two integrated grating couplers is used for coupling light into PCNC. The range of the Q factor measured was from 50,000 to 80,000. They applied a simple slope detection method with a probe laser, achieving the detection of analytes concentration by monitoring resonance wavelength shift, as shown in Figure 7c. The response of the nanobeam to MeS was 80 pm/ppm and the detection limit is 1.5 ppb, one order of magnitude lower than the value obtained by other chemical sensors previously [104]. The work demonstrates a versatile and highly integrated nanophotonic sensing platform that can be extended to detect other gas- and liquid-phase substances, simply by utilizing appropriate coating materials to achieve selective functionalization.

## 4. Nanofabrication and Coupling Techniques

Integrated optic devices have been extensively investigated to conform the trend of miniaturization and intellectualization due to the rapid advances of nano-manufacture technology [87]. The fabrication process of PCNC devices is shown in Figure 8. The two major steps are the mask process and the dry-etching process. The mask process mainly includes deep ultraviolet (DUV) lithography and electron beam (E-beam) lithography, and the dry-etching process mainly includes inductively coupled plasma (ICP) etching, electron cyclotron resonance (ECR) plasma etching and reactive ion etching (RIE). Moreover, these fabrication processes have been applied in fabrication of PCNCs. For example, Yang et al. fabricated nanoslotted parallel quadrabeam PhC cavity by using E-beam lithography and RIE [54]. Quan et al. achieved the fabrication of PhC sensor chips with DUV lithography [81]. By using E-beam lithography and ICP etching, Xu et al. fabricated a slotted PCNC with parabolic modulated width stack [66].

In order to obtain ultra-high Q, other materials have been experimentally used for the fabrication of PCNC devices, such as polymers, aluminum nitride and silicon nitride et al. In the year of 2011, Quan et al. fabricated a high Q (Q = 36,000) polymeric PCNCs with two polymers [50]. In the same year, Khan et al. fabricated and characterized silicon nitride PCNCs at visible wavelength with a high Q of ~55,000 [105]. Later, Pernice et al. presented the fabrication and characterization of a high Q aluminum nitride PCNC [106]. They obtained Q of 146,000 by using PCNCs coupled to integrated optical circuits.

Except for the nanofabrication techniques, couplers are highly desirable in the development of PCNC devices. Traditionally, several coupling methods include end-coupling, grating coupling and micro-nano fiber coupling, have been extensively used for coupling light in and out of PCNCs. For example, Quan et al. fabricated polymeric PCNCs and used tapered fiber to couple light into cavities [61], as shown in Figure 9a. Dong et al. proposed a 1D PhC low-index-mode nanobeam cavity and two grating couplers were fabricated for coupling light in/out to the cavity [107], as shown in Figure 9b. As shown in Figure 9c, Huang et al. designed PCNCs based on hetero optomechanical crystals and used a tapered and lensed fiber to couple light in/out of cavities [108].

In summary, the development of nanofabrication techniques promotes the investigation of PCNCs with a view to miniaturization and intellectualization, as well as the development of integrated optical devices. Moreover, coupling techniques are the key factors to solve the issue of optical transmission loss, playing an important part in sensing applications of PCNCs. Therefore, nanofabrication and coupling techniques need to be considered in the development of PCNC sensors. In the future, the micro-nano optical sensors can be used in internet of things (IoT) and intelligent optical network. The advanced and unique performance of the micro-nano optical sensors will play an important role in the future development of converged, collaborative and co-automatic optical networks [109].

## 5. Conclusions

In this article, we have introduced the basic properties and sensing mechanism of 1D-PCNCs. We reviewed sensing applications of 1D-PCNC sensors including ultra-high FOM RI sensing, single nanoparticle and label-free molecule detection, integrated sensor arrays for multiplexed sensing and other sensing applications. In addition, the nanofabrication and coupling techniques of 1D-PCNC sensors were reviewed. Unambiguously, the 1D-PCNC will continue to generate new opportunities for nanoscale optical sensing considering new paradigms about structures and functional materials. We believe that the 1D-PCNC is potentially a promising platform for biosensing and other lab-on-a-chip applications.

## Figures and Tables

**Figure 1 micromachines-11-00072-f001:**
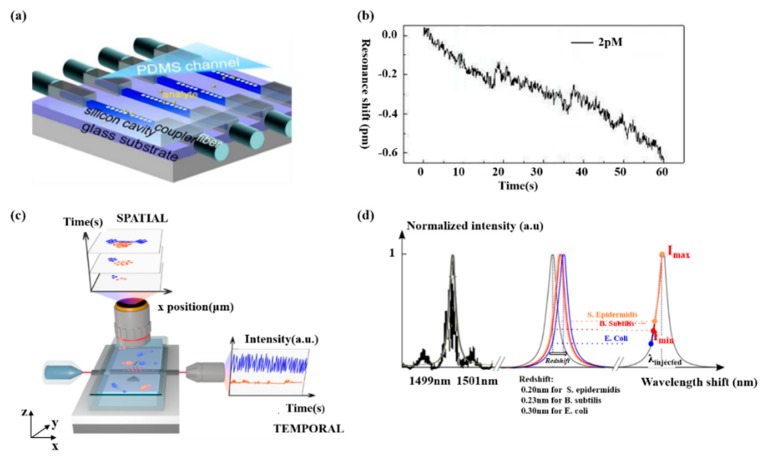
(**a**) The proposed the sensor chip, including nanobeam cavities, waveguides and PDMS (polydimethylsiloxane) channel. (**b**) Real-time response of 2 pM streptavidin phosphate buffered saline (PBS). (**a**,**b**) Reproduced with permission [21]. Copyright 2013, Optical Society of America. (**c**) Schematic of fluidic system. The blue represents E. coli and the orange is S. epidermidis. (**d**) Redshift observed in the photonic crystal nanobeam cavity (PCNC) in the presence of three molecules. (**c**,**d**) Reproduced with permission [80]. Copyright 2016, Author(s).

**Figure 2 micromachines-11-00072-f002:**
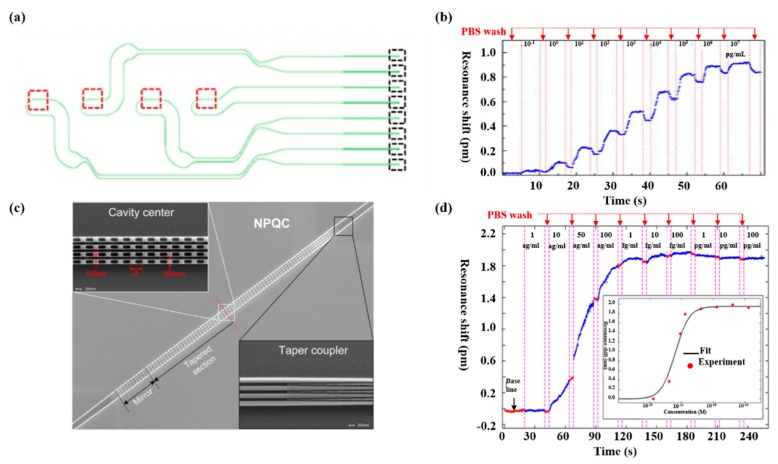
(**a**) Schematic of photonic circuits. The green lines, red boxes and black boxes represent waveguides, PCNCs and grating couplers, respectively. (**b**) Resonance shift measured in real time with different concentration of carcinoembryonic antigen (CEA). Reproduced with permission [81]. Copyright 2013, Optical Society of America. (**c**) Scanning electron microscope (SEM) image of proposed nanoslotted parallel quadrabeam PhC cavity (NPQC). (**d**) Real-time resonance shift monitored with streptavidin/biotin binding event. (**c**,**d**) Reproduced with permission [54]. Copyright 2014, AIP Publishing LLC.

**Figure 3 micromachines-11-00072-f003:**
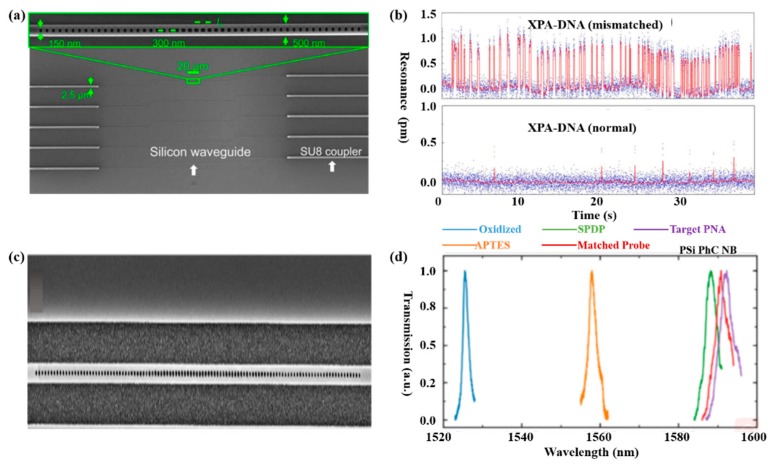
(**a**) SEM image of the silicon photonic chip consists of PCNCs (zoomed SEM inset), waveguides for the connection of nanobeam cavities to the edge for input/output coupling. (**b**) Binding dynamics of the mismatched and normal between dsDNA and xeroderma pigmentosum gene-enconded protein (XPA). (**a**,**b**) Reproduced with permission [82]. Copyright 2017, The authors. (**c**) SEM image of PCNCs patterned on porous silicon. (**d**) Transmission spectra of porous silicon PCNC, achieving the detection of five biomolecules. (**c**,**d**) Reproduced with permission [83]. Copyright 2019, Optical Society of America.

**Figure 4 micromachines-11-00072-f004:**
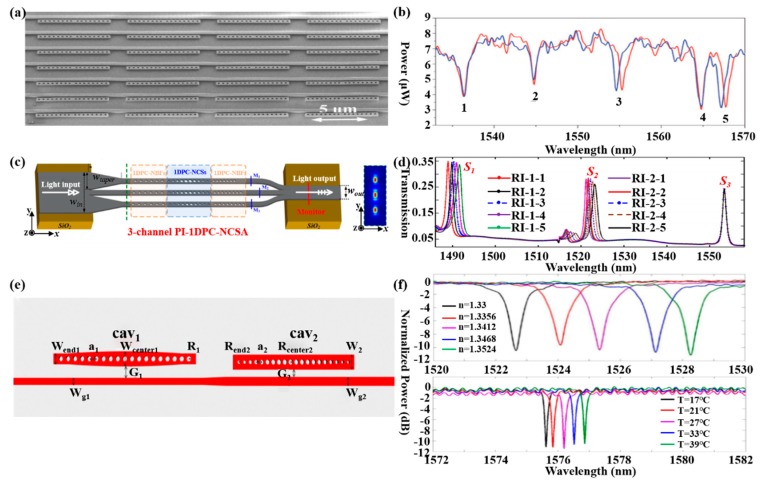
(**a**) SEM of a nanoscale optofluidic sensor array (NOSA). Reproduced with permission [84]. Copyright 2008, Optical Society of America. (**b**) Transmission spectra of resonators, achieving the detection of five molecules. Reproduced with permission [85]. Copyright 2009, The Royal Society of Chemistry. (**c**) Schematic of the proposed 3-channel parallel integrated sensor. (**d**) Transmission spectra monitored with three parallel sensing channels. The refractive index S1 and S2 change independently while S3 is not. (**c**,**d**) Reproduced with permission [86]. Copyright 2016, Optical Society of America. (**e**) Schematic of the proposed cascaded side-coupled PCNCs. (**f**) Normalized transmission spectra of cav_1_ with different glucose solutions and cav_2_ with different temperature. (**e**,**f**) Reproduced with permission [87]. Copyright 2017, Optical Society of America.

**Figure 5 micromachines-11-00072-f005:**
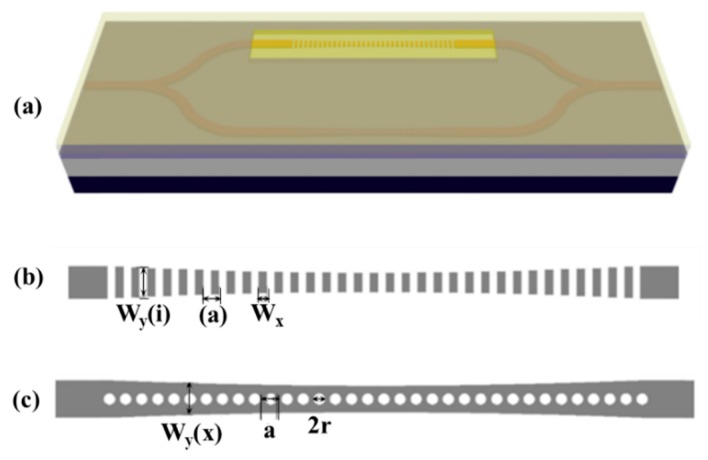
(**a**) Two cascaded PCNCs for high S temperature sensing. The PCNCs based on (**b**) stack width modulated and (**c**) parabolic-beam. (a–c) Reproduced with permission [93]. Copyright 2016, Optical Society of America.

**Figure 6 micromachines-11-00072-f006:**
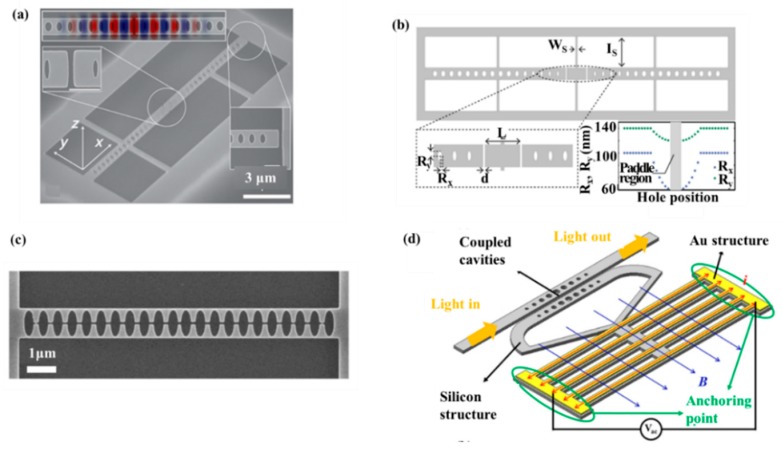
(**a**) SEM of a split-beam nanocavity. Reproduced with permission [94]. Copyright, 2014, American Physical Society. (**b**) Schematic of the PhC paddle nanocavity. Reproduced with permission [95]. Copyright, 2015, Optical Society of America. (**c**) Resonances and geometry of the sliced nanobeam. Reproduced with permission [99]. Copyright, 2015, Springer Nature. (**d**) Demonstrating of the magnetic field sensor and its working theory. Reproduced with permission [103] Copyright 2017 Author(s).

**Figure 7 micromachines-11-00072-f007:**
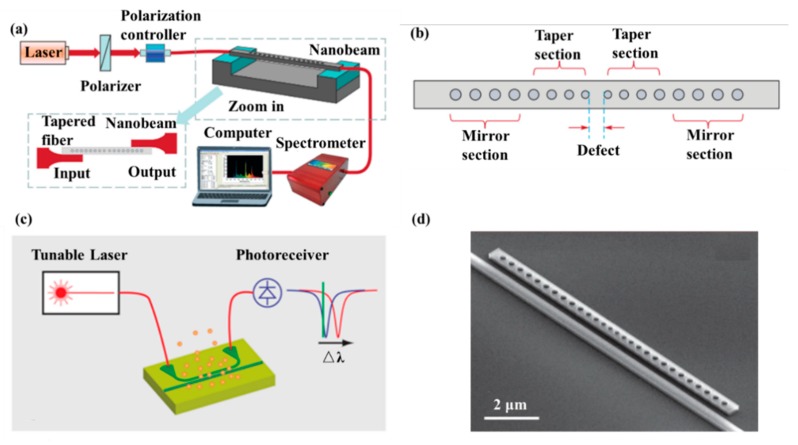
(**a**) Setup of optical gas sensor based on PCNC. (**b**) Schematic diagram of PCNC. (**a**,**b**) Reproduced with permission [103]. Copyright, 2012, Astro Ltd. (**c**) Diagram of the measurement system based on the simple slope detection method. (**d**) A PCNC coupled to a bus waveguide. (**c**,**d**) Reproduced with permission [104]. Copyright 2014, American Chemical Society.

**Figure 8 micromachines-11-00072-f008:**
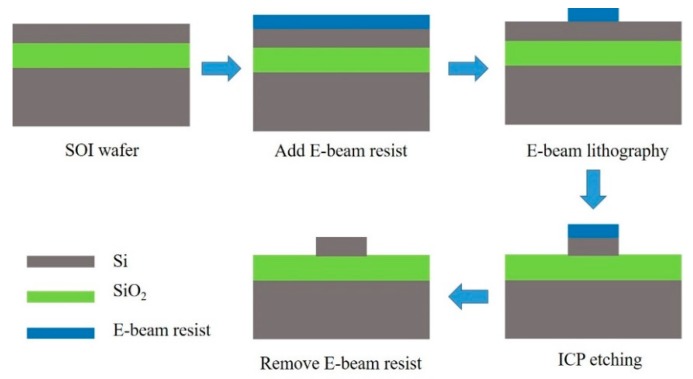
Illustration of fabrication process of PCNCs.

**Figure 9 micromachines-11-00072-f009:**
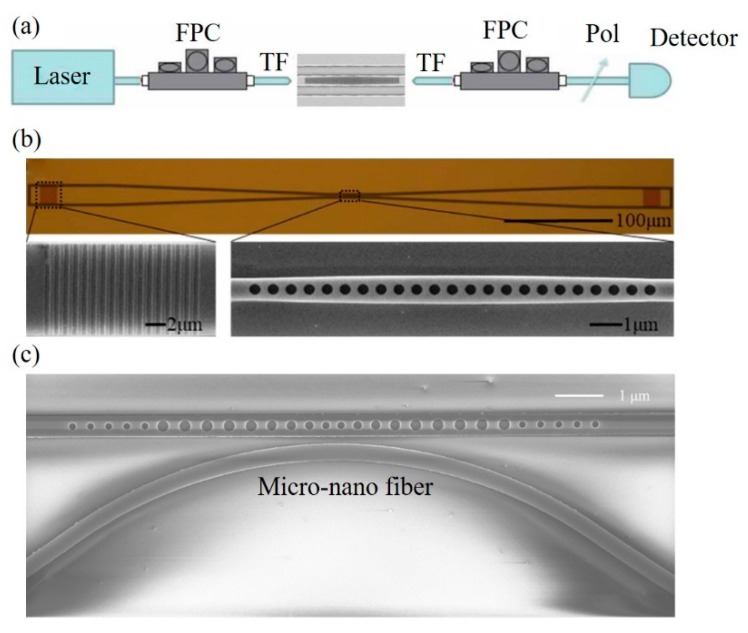
(**a**) Schematics of measurement setup. FPC: fiber controller cavity, TF: tapered fiber, Pol: inline polarizer. Reproduced with permission [61]. Copyright, 2011, Optical Society of America. (**b**) Top: Microscopy image of the proposed sensor and coupler. Bottom: SEM of grating coupler (left) and PhC cavity (right). Reproduced with permission [107]. Copyright 2019, Optical Society of America. (**c**) SEM of 1D optomechanical crystal cavity with a micro-nano fiber coupler. Reproduced with permission [108]. Copyright 2016, SPIE.

**Table 1 micromachines-11-00072-t001:** Q, V, S and FOM (figure of merit) compared with different structure.

Classification	Structure	QualityFactor (Q)	ModeVolume (V)	Sensitivity(nm/RIU)	FOM	Ref
**High-Q** **(Q > 10^4^)** **Low-S** **(S < 300)**	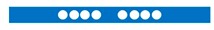	20,000	--	200	2580	[47]
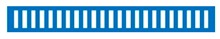	16,000	2.0(*λ*/n)^3^	--	--	[48]
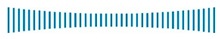	1.74 × 10^7^	1.48(*λ*/n_Si_)^3^	269	4587	[49]
**Low-Q** **(Q < 10^4^)** **High-S** **(S > 300)**	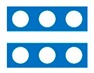	700	--	900	419	[53]
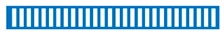	--	--	321	--	[59]
**High-Q** **(Q > 10^4^) and** **High-S** **(S > 300)**	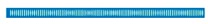	36,000	--	386	9190	[61]
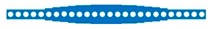	5.16 × 10^6^	2.18(*λ*/n_Si_)^3^	537.8	~4000	[50]
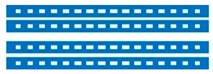	>10^7^	--	>800	~5000	[62]
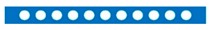	>23,300	--	631	>9500	[63]
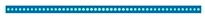	2.1 × 10^5^	--	736.8	1.13 × 10^5^	[64]
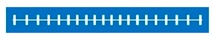	5.5 × 10^5^	0.03(*λ*/n_air_)^3^	835	2.92 × 10^5^	[65]
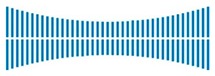	3.73 × 10^6^	0.217(*λ*/n_c_)^3^	437	1.5 × 10^6^	[66]
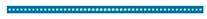	1.1 × 10^7^	2.93(*λ*_SiO2_)^3^	563.6	4.31 × 10^6^	[64]
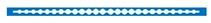	2.67 × 10^7^	0.01(*λ*/n_air_)^3^	~750	1.31 × 10^7^	[67]

**Table 2 micromachines-11-00072-t002:** Examples for single nanoparticle detection based on photonic crystal nanobeam cavity (PCNC).

Structure	Q	Radius	Ref
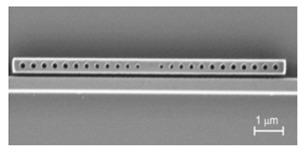	2000	55 nm	[76]
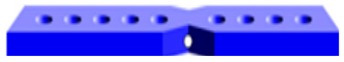	1837	50 nm	[77]
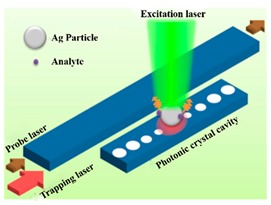	1500	40 nm	[74]
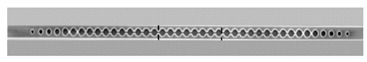	1.2 × 10^4^	20 nm	[58]
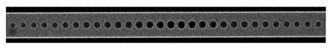	1.2 × 10^5^	12.5 nm	[21]
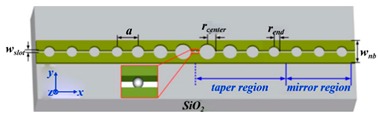	--	10 nm	[78]
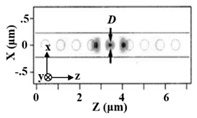	190	5 nm	[79]
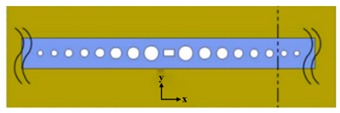	6.08 × 10^6^	1 nm	[55]
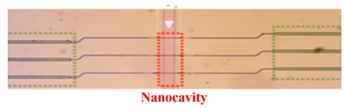	2.5 × 10^5^	0.9 nm	[44]

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
