# Peer review of "Photonic Crystal Nanobeam Cavities for Nanoscale Optical Sensing: A Review"

_micromachines, 2020, doi:10.3390/mi11010072_

Round 1

Reviewer 1 Report

This article gives a comprehensive for the research works of PCNC sensors. Most of the critical works in this field have been listed out and well organized. Despite, the article still needs to be improved in several aspects. 1. The research works should not just be reviewed by listing out the methods and results, but also be well commented. For example, in section 3.1, all these structure designs should be classified into several categories. With the progress of achieving high FOM, what are the key points of improvements in the structure design? What are the research directions people mainly focusing on currently for further progress and what are the authors’ opinions for the future of this field? Similar questions should be addressed in other sections 2. The authors pointed out that the linewidth broadening effect can be used for sensing. However, no previous work has been listed or discussed in the following parts. All the single particle and biomolecule sensing are achieved by wavelength shift. More works on the broadening effect sensing should be reviewed. 3. Among several types of optical sensors (nano-waveguide, WGM, PhC, etc.), what are the benefits and challenging points of PCNC sensors? Different types of sensors are suitable for different particles types and environmental conditions. The review should express deeper in this aspect and show how the research field tried to overcome these challenges. 4. Some typo errors need to be corrected. For example, in line 133, “the greater the Q, the light mode should be confined to the waveguide medium” change to “the greater the Q, the light mode should be more confined to the waveguide medium”.

Reviewer 2 Report

This authors present a comprehensive review on the sensing functions for photonic crystal nanobeam cavities. In general this is an interesting emerging technique for ultra-low volume sensing that worth to review. The authors have provided an useful comparison among different realizations & performances for the previously reported systems. The structure of the review is also well-organized, I therefore suggest the publication of this work in Micromachines, given the following points are properly addressed.

1. The performance of the nanobeam cavity strongly depends on the advances in nanofabrication. It would be very useful to add one more sections reviewing on the available fabrication techniques of the nanocavity, as well as the formed materials.

2. It would also be useful to mention the useful techniques to couple light in and out of the cavity, as well as the demonstrated efficiencies, since this is one of the key for the sensing applications.

3. Since the authors mentioned the function of particle detection & environmental monitoring, the following reference is worth to be cited, either in the introduction section mentioning other available techniques, or in the section of single particle detection.

[Opt. Express 27(24), 34496-34504, 2019]
